# Identifying trends in reporting on the ethical treatment of insects in research

Craig D. Perl[ID][1,2], Cat Kissinger[3], R Keating Godfrey[4], Enrique M. Castillo[3], Bob Fischer[3], Meghan Barrett[ID][5]*

1 Insect Welfare Research Society, Indianapolis, Indiana, United States of America, 2 Institute of Aquaculture, University of Stirling, Stirling, United Kingdom, 3 Department of Philosophy, Texas State University, San Marcos, Texas, United States of America, 4 Department of Biological Sciences, Dartmouth College, Hanover, New Hampshire, United States of America, 5 Department of Biology, Indiana University Indianapolis, Indianapolis, Indiana, United States of America

* meghbarr@iu.edu

## Abstract

Transparent reporting on the ethical treatment of research animals (e.g., implementation of the 3Rs, replace, reduce, and refine) is recommended when publishing in peer-reviewed literature. This is meant to foster public trust, safeguard animal welfare, and generate reproducible science. However, entomologists are not expected to engage in such reporting, as their research is not subject to legislated ethical review. Recently, however, entomologists have reported increased concern about the ethical treatment of insects in research, and associated reproducibility and public trust issues. To what degree are these increasing concerns reflected in changes in practices? We surveyed 15 high-impact journals that publish on insects over 20 years to collect data on reporting related to the ethical treatment of insects in research, including animal reduction methods, analgesics/anesthesia statements, and information regarding sacrifice. Out of 1359 sampled papers, no studies reported any methods to reduce animal use. Over 20 years, we found an increase in the proportion of papers reporting insect death and a decrease in the papers reporting significant invasive handling. 84% of papers with significant animal handling or death did not report the use of any anesthetics. We also found an increase in animal-treatment-specific ethics statements (from 0% to 8%), largely driven by the journal *Animal Behaviour*. We end by 1) making recommendations for entomologists looking to improve their reporting practices and 2) providing tools to improve transparent reporting of information related to the ethical treatment of insects in research.

## Introduction

The ethical treatment of animals used in research is a cornerstone of generating reproducible science, safeguarding animal welfare, and fostering public trust.

**Data availability statement:** Data is publicly available with all associated metadata on the Open Science Framework Digital Repository (https://doi.org/10.17605/OSF.IO/HN7JC).

**Funding:** CDP is employed by the Insect Welfare Research Society. This research was thus made possible thanks to the financial contributions of many donors enabling the work conducted by the Insect Welfare Research Society. None of the other authors were funded to work on this project.

**Competing interests:** MB and BF report a relationship with the Insect Welfare Research Society that includes: board of advisors (unpaid). CDP reports a relationship with the Insect Welfare Research Society that includes: employment. This does not alter our adherence to PLOS ONE policies on sharing data and materials.

Transparent reporting on the practices used during animal experiments is thus recommended when publishing peer-reviewed literature [1]. Historically, however, similar norms have not been considered best practice for animals excluded from legislated ethical oversight [2,3], such as insects ([4] but see [5]). In these cases, disciplines like entomology have developed permissive research and reporting norms for animal use relative to those that are standard for other animals. Where these permissive norms create risks for 1) public trust, 2) animal welfare, or 3) reproducible science, entomologists may wish to review their norms and establish new best practices for the field.

There are reasons to think that permissive norms indeed create such risks. For instance, recent research shows that public trust in invertebrate research declines without ethical oversight [6], suggesting that entomologists may also need to transparently report on ethical practices used in research like vertebrate biologists. Further, public opinion on insect sentience (i.e., the capacity for valanced states like pain) may be shifting [7], at least in Western nations. 47.2% of American adults believed insects used as livestock feed could feel pain, with only 18.9% uncertain; further, younger respondents were more likely to be concerned about the ethical treatment of insects [8]. Numbers may be even higher for more charismatic animals commonly used in the lab, such as bees: a survey of American adults found differences in the public's belief in the plausibility of insect pain was highest for bees (65% yes, 27% unsure), compared to ants (56% yes, 32% unsure) or termites (52% yes, 35% unsure; [9]). Recent examples of public outcry about lethal insect collecting studies (e.g., [10]) suggests that the public is becoming more concerned about ethical treatment of insect research subjects, which matches growing attention to this issue by researchers as well [11–13].

Although there is no scientific consensus on insect sentience, a recent review [5] suggests there is enough neurobiological and behavioural evidence to warrant precautionary approaches towards their welfare in research until more data can be collected [12,14–18]. Indeed, such a precautionary approach has even been recommended by professional entomologists who are skeptical about insect sentience [4]. Reporting on ethical animal use practices is itself a precautionary measure, encouraging animal researchers to articulate and refine their practices to reduce avoidable harm [19].

Finally, reporting on ethical animal use is a critical part of reproducible science wherever either affective states or physiological/behavioural changes due to stressors may be expected to confound or otherwise alter study results (see examples from the vertebrate literature: [20–22]). Just as in vertebrates, studies in insects suggest that many welfare-relevant practices can influence study results. For example, some methods of anesthesia (e.g., $CO_2$) can impact behaviour for days after the anesthetic was applied in *Drosophila melanogaster,* while others (halogenated ethers) do not have the same behavioural effects [23]. A lack of reporting on the anesthetics used, the duration, and the dosage applied can thus represent a reproducibility concern in the literature that better reporting on animal welfare would simultaneously address [24].

A recent survey of entomologists suggests there is growing concern about insect welfare based on the considerations just surveyed—i.e., preserving public trust,

the intrinsic value of animal welfare itself, and generating reproducible science [25]. However, it is unclear whether and how this increase in concern is reflected in practice. One valuable public metric is increased reporting of welfare-relevant information in entomological publications. To determine trends in historical reporting of the ethical treatment of insects, we surveyed five papers per year from the past 20 years (2003–2022) that use live insects from 15 peer-reviewed journals with a range of impact factors, subdisciplinary breadth, and taxonomic focus.

We report on trends in the data related to two components of the standard 3Rs framework for ethical animal use in research: reduction (using fewer animals when possible) and refinement (minimizing harm to living animals when possible; [26]). This framework has recently been recommended as a best practice for use with invertebrates [15,27]. We thus looked for reports that authors were reducing animal use to the minimum number required (e.g., via statistical methods for determining appropriate power), or refining their practices via the use of anesthetics/analgesics prior to significant handling or non-instantaneous sacrifice. Further, we discuss the prevalence of explicitly-labeled ethics statements. Finally, we make recommendations for how entomologists could improve transparent reporting of information related to the ethical treatment of their research subjects in the peer-reviewed literature, with the goal of addressing concerns about public trust, welfare, and reproducibility.

## Methods

### Journal and article selection

We used the Scopus CiteScore rankings, as of January 2023, to determine the top 40 entomology journals. We excluded any journals from the list that, by name, focus mostly on review papers, non-animals (e.g., plants, fungi, etc.), or non-insect animals (e.g., ticks, etc.). From the remaining 26 journals, we selected 9 that represented a broad swath of applied and fundamental entomological disciplines, to capture a range of methods and sub-disciplinary practices. Applied insect-focused journals included: *Journal of Pest Science, Insect Conservation and Diversity, Medical and Veterinary Entomology*, and *Journal of Insects as Food and Feed.* Fundamental insect-focused journals included: *Insect Biochemistry and Molecular Biology, Insect Science, Journal of Insect Physiology, Ecological Entomology*, and *Arthropod Structure and Development.* We selected an additional six, high-impact journals that publish fundamental papers from a variety of subdisciplines, but that are not taxonomically focused on insects though frequently featuring insect work: *Nature, PNAS, Journal of Experimental Biology, Animal Behavior, eLife*, and *Proceedings of the Royal Society B.* These journals were selected to better understand if publishing alongside vertebrate articles would shape entomological ethics reporting practices.

Five articles were randomly selected from each journal for each year they were in operation between 2003 and 2022 (n = 1,359). Articles that did not report primary research on insect specimens were not used and were replaced by another randomly selected article from that journal in that year, unless no additional articles that met these criteria could be found. We did not include studies conducted on previously-dead insects (e.g., a study just on museum samples).

### Article assessment and searches

The methods of each article were read and assessed to determine 1) if any insects were significantly handled during the experiment (e.g., restraint, dissection) and 2) if any insects died during the experiment (e.g., whole-body sampling for DNA) or as a direct result of experimental procedures (e.g., electrophysiology). Other papers likely involved insect death as part of their ongoing experimental procedure but animal sacrifice may not have been an explicit part of their protocol (e.g., rearing of fruit flies for behavioural investigations). These were not recorded as having involved animal sacrifice. The methods of each paper were also searched for reporting of any *a priori* statistical analyses for assessing the number of animals needed to complete an experiment (e.g., power analyses).

Further, articles were searched automatically for the following keywords: 'anesth*', 'anaesth*', 'analg*', 'welfare', 'ethic*', 'handling', 'manipulation', 'cold', 'freeze*', 'froze*', 'ice', 'liquid (nitrogen)', 'carbon (dioxide)', 'CO2', 'ether', '*flurane', 'euthan*', 'sacrific*', 'ethanol', 'morphine', 'opioid', 'GABA', 'gaba*', 'ɣ-aminobutyric acid', 'potassium chloride', 'NSAID', 'antidepressant'. When these terms resulted in a 'hit', the area of the paper containing the search term was read to determine if the hit was welfare relevant (e.g., a sacrifice or analgesia/anaesthesia statement, or something else). For welfare-relevant hits, the method of anesthesia/sacrifice was recorded, the location of the hit within the paper (methods, or elsewhere), and if the hit was part of an explicit 'ethics' statement or only in the general methodology. Only the main materials (i.e., not supplemental files) of publications were searched for these terms.

### Statistical analyses

All statistical analyses were conducted using R v. 4.4.0 (R Core Team, 2022). Linear regressions, GLMs, Pearson-product moment correlations, *t*-tests, ANOVAs and Tukey Honest Significant Difference tests were implemented using the R base package. Alpha was set to 0.05. Data and code are publicly available with all associated metadata via the Open Science Foundation (doi.org/10.17605/OSF.IO/HN7JC).

## Results

Throughout the results, 'ethics statement' refers to any ethics statement appearing in a paper that was welfare-relevant (e.g., statements about permitting, or that say 'no ethics approval required for insects', were excluded). 'Anesthetic/analgesic statement' refers to any mention in the paper of an anesthetic or analgesic being employed during the study, whether that information is reported for ethical or methodological reasons. 'Sacrifice statement' refers to any mention in the paper of an animal being sacrificed, whether that information is reported for ethical or methodological reasons.

### Description of sacrifice and anesthetic usage in papers

No papers in our sample reported the use of any *a priori* methods (e.g., power analyses) for reducing the number of animals used in the experiment.

A mean (± standard deviation) of 77.17±4.79% of papers across all years contained insect death during or resulting from studies (range: 63.33–82.35%; Table 1). 59.90%±7.18% of papers contained insect death with no significant handling (48.33–70.77%) while 17.11%±5.62% of papers contained significant handling prior to death (range: 7.69–27.12%). 10.27%±5.04% of papers contained significant handling not followed by death (range: 4.00–23.33%). Finally, 12.48%±3.25% of papers contained neither significant handling nor death resulting from the study (range: 5.08–18.33%).

The most common anesthetics/analgesics (Table 2) used prior to significant handling or death (and found as a result of our search terms) were freezing/chilling (n=126 reports) and carbon dioxide (n=80). Less commonly reported methods were chloroform (n=1), nitrogen/nitrogen monoxide (n=2), ether (n=3), isoflurane (n=1). 83.68% of papers that reported significant handling or death did not report the use of any anesthetics/analgesics prior to the event (Table 2).

When sacrifice methods were captured by our automated methods, cold/freezing/$LN_2$/dry ice (n = 120) and dissection (n = 48) were the most commonly reported (Table 3). Less commonly reported methods were immersion in ethanol/solvent/fixative (n = 24) and homogenisation (n = 15), among others. 79 studies that our automated methods highlighted as containing an animal sacrifice method did not report which method was used to kill their animals (38.05%).

1051 total papers containing insect death (with 1161 instances of sacrifice, as some papers used more than one method of sacrifice) were manually identified, with 870 instances of insect death that were not recognized by our automated methods. Of the papers where the sacrifice method was captured manually (Table 4), the most common methods were dissection (n=304) or cold/freezing/$LN_2$/dry ice (n=137). The largest contingent of papers (Table 4; n=307), did not explicitly state any sacrifice method but on examination of the methods, death could be inferred with certainty. The

**Table 1. Number of papers sampled and their respective counts and proportions of animal sacrifice and invasive handling per year.**

| Publication year | Papers sampled | All papers containing sacrifice | | Papers containing sacrifice without invasive handling | | Papers containing both invasive handling and sacrifice | | Papers containing invasive handling without sacrifice | | Papers containing neither sacrifice or invasive handling | |
|---|---|---|---|---|---|---|---|---|---|---|---|
| | n | n | % | n | % | n | % | n | % | n | % |
| 2003 | 59 | 48 | 81.36 | 32 | 54.24 | 16 | 27.12 | 8 | 13.56 | 3 | 5.08 |
| 2004 | 60 | 42 | 70.00 | 29 | 48.33 | 13 | 21.67 | 11 | 18.33 | 7 | 11.67 |
| 2005 | 60 | 38 | 63.33 | 29 | 48.33 | 9 | 15 | 14 | 23.33 | 8 | 13.33 |
| 2006 | 61 | 45 | 73.77 | 37 | 60.66 | 8 | 13.11 | 11 | 18.03 | 5 | 8.2 |
| 2007 | 59 | 46 | 77.97 | 41 | 69.49 | 5 | 8.47 | 6 | 10.17 | 7 | 11.86 |
| 2008 | 65 | 51 | 78.46 | 46 | 70.77 | 5 | 7.69 | 4 | 6.15 | 10 | 15.38 |
| 2009 | 65 | 50 | 76.92 | 37 | 56.92 | 13 | 20 | 9 | 13.85 | 6 | 9.23 |
| 2010 | 69 | 54 | 78.26 | 45 | 65.22 | 9 | 13.04 | 7 | 10.14 | 7 | 10.14 |
| 2011 | 60 | 43 | 71.67 | 36 | 60.00 | 6 | 10 | 6 | 10 | 11 | 18.33 |
| 2012 | 62 | 51 | 82.26 | 41 | 66.13 | 10 | 16.13 | 4 | 6.45 | 7 | 11.29 |
| 2013 | 68 | 56 | 82.35 | 46 | 67.65 | 10 | 14.71 | 5 | 7.35 | 7 | 10.29 |
| 2014 | 68 | 55 | 80.88 | 44 | 64.71 | 10 | 14.71 | 4 | 5.88 | 9 | 13.24 |
| 2015 | 77 | 56 | 72.73 | 39 | 50.65 | 17 | 22.08 | 10 | 12.99 | 11 | 14.29 |
| 2016 | 75 | 60 | 80 | 50 | 66.67 | 10 | 13.33 | 7 | 9.33 | 8 | 10.67 |
| 2017 | 75 | 59 | 78.67 | 45 | 60.00 | 14 | 18.67 | 3 | 4 | 13 | 17.33 |
| 2018 | 75 | 58 | 77.33 | 38 | 50.67 | 20 | 26.67 | 5 | 6.67 | 12 | 16 |
| 2019 | 74 | 57 | 77.03 | 41 | 55.41 | 16 | 21.62 | 7 | 9.46 | 10 | 13.51 |
| 2020 | 73 | 57 | 78.08 | 40 | 54.79 | 17 | 23.29 | 4 | 5.48 | 12 | 16.44 |
| 2021 | 75 | 61 | 81.33 | 50 | 66.67 | 11 | 14.67 | 5 | 6.67 | 9 | 12 |
| 2022 | 79 | 64 | 81.01 | 48 | 60.76 | 16 | 20.25 | 6 | 7.59 | 9 | 11.39 |

**Table 2. List of anaesthetics reported in papers with invasive handling or insect sacrifice.**

| Anaesthetic | n | % |
|---|---|---|
| chloroform | 1 | 0.08 |
| isoflurane | 1 | 0.08 |
| nitrogen | 1 | 0.08 |
| nitrogen monoxide | 1 | 0.08 |
| ether | 3 | 0.25 |
| unspecified method* | 5 | 0.42 |
| freezing | 21 | 1.77 |
| $CO_2$ | 80 | 6.73 |
| fhilling | 105 | 8.83 |
| not stated** | 995 | 83.68 |

*"Unspecified" refers to papers that mentioned the word "anaesthetic" or "anaesthesia" but did not specify which anaesthesia method was used.

**"Not stated" refers to those papers with invasive handling or insect sacrifice but without any reported anaesthetic or analgesic information

**Table 3. Sacrifice methods reported in papers extracted by our automated process. "Not stated" category refers to those papers identified as having an insect die as a result of their methods but that did not describe the process that resulted in death.**

| Sacrifice method | n | % |
| --- | --- | --- |
| boiled | 1 | 0.49 |
| cold | 1 | 0.49 |
| diethyl ether | 1 | 0.49 |
| ether | 1 | 0.49 |
| glycerol | 1 | 0.49 |
| irradiated | 1 | 0.49 |
| $CO_2$ | 2 | 0.97 |
| pesticide | 2 | 0.97 |
| dry ice | 3 | 1.46 |
| fixative | 3 | 1.46 |
| solvent | 6 | 2.91 |
| ethanol | 11 | 5.34 |
| homogenized | 15 | 7.28 |
| liquid nitrogen | 33 | 16.02 |
| dissected | 48 | 23.3 |
| not stated | 79 | 38.35 |
| frozen | 83 | 40.29 |

most common experimental procedures from which insect death could be inferred (S1 Table) were nucleic acid extraction (n = 119) and electrophysiology (n = 28).

Of all the papers reporting dissection as a sacrifice method, 52 (17.21%) reported the use of an anaesthetic or analgesic prior to the dissection, with chilling/freezing (n = 33) and carbon dioxide (n = 18) the most commonly employed methods (S2 Table).

## Trends in reporting over time

The percentage of studies that contained insect death (with or without invasive handling) increased from 2003 to 2022 (Fig 1; linear regression; $t_{20,18} = 2.33$, $p = 0.031$, $R^2 = 0.23$) while the percentage of studies that contained just invasive handling decreased (linear regression; $t_{20,18} = 4.00$, $P = 0.001$, $R^2 = 0.44$). There were no other significant changes over time in the instances of death with no invasive handling, death preceded by invasive handling, or studies with neither handling nor death (linear regression; $t_{20,18} < 2.06$, $p > 0.05$). There was no significant relationship over time in the percentage of studies that contained dissection with no accompanying anaesthesia or analgesia statement (linear regression; $t_{20,18} = 1.62$, $p = 0.12$).

The percentage of studies that contained an ethics statement increased over time from 0.00% in 2003 to 7.59% in 2022 (maximum = 9.59% in 2020) (Fig 2; linear regression, $t_{20,18} = 4.29$, $p < 0.001$, $R^2 = 0.48$).

## Trends in reporting by journal

Comparing prevalence of ethics statements among journals (Fig 3) indicated that the aforementioned increase in ethics statements over time was largely driven by a single publication: *Animal Behaviour.* However, even when excluding this journal from the analysis, there was still a slight but significant increase in the use of ethics statements across the other journals from 0.00% in 2003 to 1.35% by 2022 (Fig 2; linear regression, $t_{20,18} = 3.83$, $p = 0.001$, $R^2 = 0.42$).

**Table 4. Sacrifice methods reported in papers extracted by manual searching.**

| Sacrifice method | n | % |
|---|---|---|
| dry ice | 1 | 0.1 |
| microwave | 1 | 0.1 |
| $CO_2$ | 2 | 0.19 |
| cold | 2 | 0.19 |
| irradiated | 2 | 0.19 |
| starved | 2 | 0.19 |
| Bt toxin | 3 | 0.29 |
| heat treatment | 3 | 0.29 |
| drowned | 5 | 0.48 |
| boiled | 6 | 0.57 |
| crushed | 7 | 0.67 |
| dessicated | 8 | 0.76 |
| infection | 11 | 1.05 |
| glycol | 12 | 1.14 |
| solvent | 12 | 1.14 |
| predation-parasitism | 14 | 1.33 |
| exsanguinated | 18 | 1.71 |
| liquid nitrogen | 20 | 1.90 |
| decapitated | 25 | 2.38 |
| trap | 27 | 2.57 |
| homogenized | 30 | 2.85 |
| fixative | 32 | 3.04 |
| experimental death* | 56 | 5.33 |
| pesticide | 58 | 5.52 |
| ethanol | 78 | 7.42 |
| frozen | 115 | 10.94 |
| dissected | 304 | 28.92 |
| not stated** | 307 | 29.21 |

*The "experimental death" category refers to those papers were insects were explicitly stated to have died through the course of rearing or during development.

**The "not stated" category refers to those papers identified as having an insect die as a result of their methods but that did not explicitly describe the process that resulted in death.

*Animal Behaviour* is the only journal in our sample to currently require an ethics statement, even for studies on invertebrates. We therefore first tested for any significant differences in the proportion of papers with an ethics statement among the 14 other journals, where no ethics statement was required when working with invertebrates. There was no significant difference among them (ANOVA; $F_{13,253} = 1.29$, p=0.22). We then collapsed them into a single factor, in order to compare journals where no ethics statement is required for studies with invertebrates to journals where this is required – in this case, only *Animal Behaviour*. *Animal Behaviour* had a significantly higher mean proportion of ethics statements than the journals that did not require an ethics statement when working with invertebrates (Fig 3; $t_{19,273} = 3.38$, p-value<0.01). *Animal Behaviour* had a significantly higher mean percentage of ethics statements (mean±SE; 24.2±6.89) than the journals that did not require an ethics statement (mean±SE; 0.95±0.29) when working with invertebrates, combined ($t_{19,273} = 3.38$, p-value<0.01).

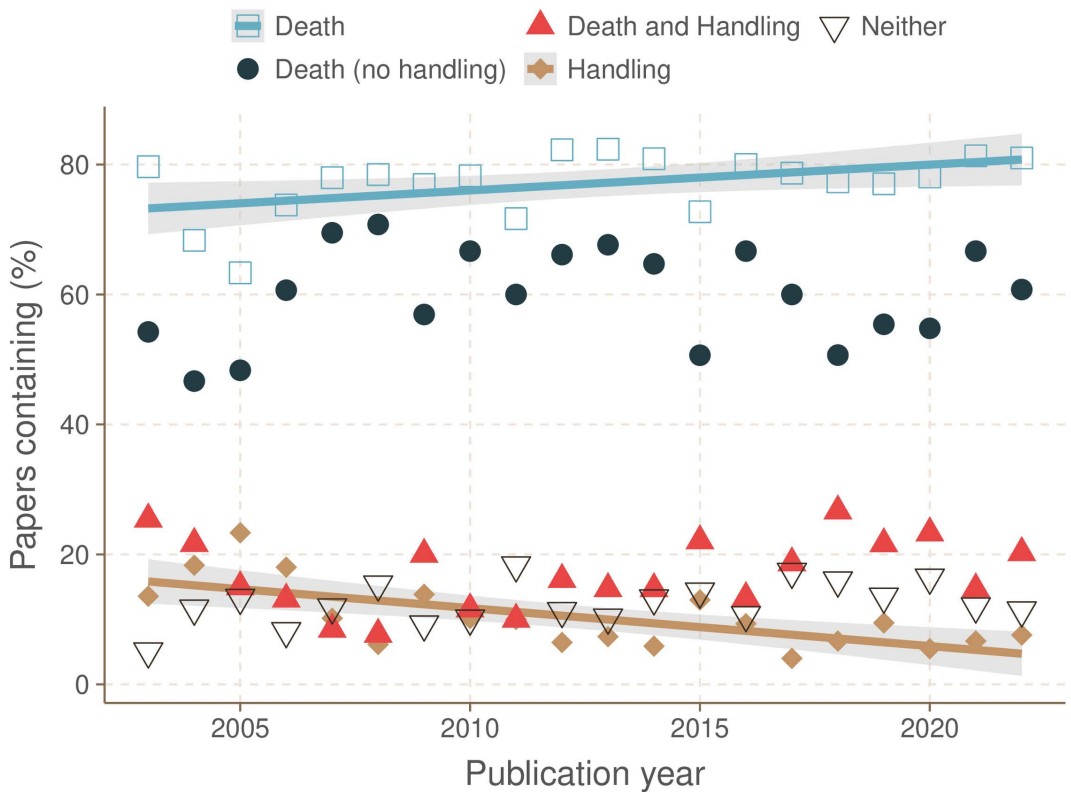

**Fig 1. The percentage of studies that resulted in insect death increased over time.** There was a significant increase in the proportion of studies containing insect death (with or without handling) over time (linear regression; $t_{20,18}=2.33$, $p=0.031$, $R^2=0.23$). There was a significant decrease in the proportion of studies reporting just invasive handling (without death) over time (linear regression; $t_{20,18}=4.00$, $p=0.001$, $R^2=0.44$). There was no change in the percentage of studies containing death only (no handling), both death and significant handling, or neither death nor significant handling (linear regression; $t_{20,18}<1.19$, $p>0.05$).

Six journals (*Animal Behaviour, eLife, Journal of Experimental Biology, Nature, Proceedings of National Academy of Sciences, Proceedings of the Royal Society B*) from our sample publish papers investigating both invertebrates and vertebrates, while the remaining nine (*Arthropod Structure and Development, Ecological Entomology, Insect Biochemistry and Molecular Biology, Insect Science, Journal of Insect Physiology, Journal of Pest Science, Medical and Veterinary Entomology, Insect Conservation and Diversity, Journal of Insects as Food and Feed*) publish almost exclusively invertebrate research. Journals that also publish vertebrate research had a significantly higher proportion of papers reporting an anaesthesia or analgesia statement compared with journals that publish only invertebrate research ([Fig 4](); *t*-test, $t_{40,38}$, 3.19, $p<0.003$). Journals that publish vertebrate research also had a significantly higher proportion of papers reporting a sacrifice statement compared with journals that publish only invertebrate research ([Fig 4](); *t*-test, $t_{40,38}=3.05$, $p<0.005$). Four of six journals (66.67%) that also publish on vertebrate research had at least one paper with an ethics statement in our sample; this was only true of four of nine journals (44.44%) that publish almost exclusively on invertebrate research ([Fig 3]()).

## Trends in reporting by Taxa

To facilitate robust statistical analysis, the seven most common orders ([S1 Fig](); Diptera, Hymenoptera, Coleoptera, Lepidoptera, Hemiptera, Orthoptera, Blattodea) within our sample were selected for analysis; all other taxa were collapsed

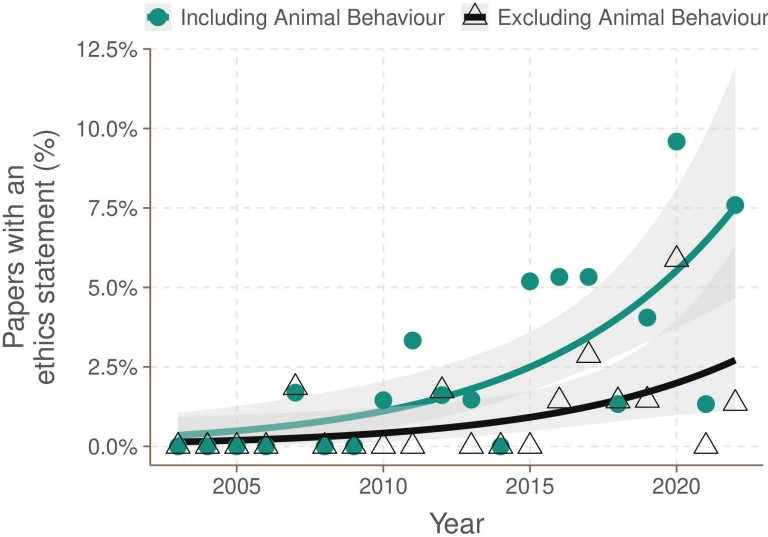

**Fig 2. The percentage of studies that contained an ethics statement increased over time, largely driven by *Animal Behaviour*.** There was a significant increase in the proportion of studies containing an ethics statement over time (green line; general linear model, quasibinomial family, $t_{20,18}=4.35$, $p<0.001$). Much of this increase in the use of ethics statements was driven by publications in *Animal Behaviour*, however even with this journal excluded there was still a significant increase from 0.00% in 2003 to 1.35% in 2022 in the proportion of studies containing an ethics statement in the other 14 journals (black line; general linear model, quasibinomial family, $t_{20,18}=2.40$, $p=0.03$).

into a single category ("Other Orders"), creating a total of eight categories. The seven most common orders accounted for 94% of the instances within the data.

There was a significant difference in the proportion of anaesthesia statements prior to invasive handling among orders (Fig 5A; ANOVA; $F_{7,8}=6.36$, $p<0.0001$). Investigations using Diptera contained a significantly greater proportion of relevant anaesthesia statements than investigations using Coleoptera (Tukey HSD; $p=0.02$), Lepidoptera (Tukey HSD; $p=0.001$), Orthoptera (Tukey HSD; $p<0.0001$) and Other Orders (Tukey HSD; $p=0.001$). Investigations using Hymenoptera had a greater proportion of relevant anaesthetic statements than those using Orthoptera (Tukey HSD; $p=0.04$). We found no significant difference in the proportion of sacrifice statements (Fig 5B; ANOVA; $F_{7,8}=0.21$, $p=0.98$), or the proportion of anaesthesia statements (Fig 5C; ANOVA; $F_{7,8}=0.86$, $p=0.54$), prior to death among orders.

## Discussion

We sampled 1359 papers that used live insects from 15 journals across a 20-year period to ascertain trends in reporting on *a priori* power analyses, anesthetic use, and sacrifice methods, as well as explicit ethics statements focused on animal welfare, across the discipline. Our study was only designed to look for specific indicators of 'reduction' and 'refinement' in studies where live animals were used (two of the 3Rs [26]); we did not attempt to determine if there were opportunities in any studies to have replaced live animals with less-sentient alternatives (such as replacing live animals with museum specimens or performing studies on cells instead of whole organisms).

### Reporting on animal reduction via a priori sample size estimation

No paper in our sample reported the use of *a priori* methods for determining sample sizes, a process that is recommended to reduce the number of animals used to the minimum necessary for a study to have appropriate power [28,29]. This practice has been shown to be essential for producing valid scientific conclusions as well as promoting animal welfare [30–32]. Estimating sample sizes is a more common practice in vertebrate research, as ethical review boards can serve as a point

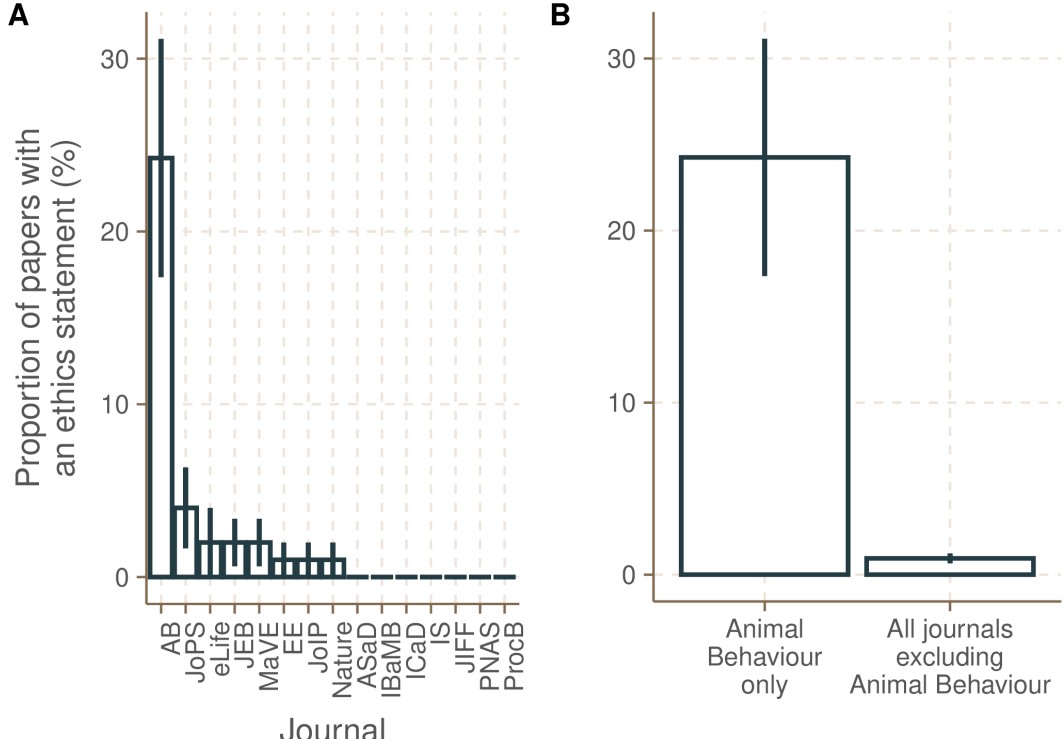

**Fig 3. The journal *Animal Behaviour* has a greater proportion of papers with an ethics statement than any other journal. (A)** None of the other 14 journals, excluding *Animal Behaviour*, have a significantly different proportion of papers with ethics statements from each other (ANOVA, $F = 1.29$, $p = 0.22$). Journal abbreviations: ASaD - *Arthropod Structure and Development*; EE – *Ecological Entomology*; IBaMB - *Insect Biochemistry and Molecular Biology*; IS – *Insect Science*; JEB – *Journal of Experimental Biology*, JoIP – *Journal of Insect Physiology*; JoPS – *Journal of Pest Science*; MaVE – *Medical and Veterinary Entomology*; *Nature* – Nature; PNAS – *Proceedings of the National Academy of Sciences*; ProcB: *Proceedings of the Royal Society B*; ICaD - *Insect Conservation and Diversity*; JIFF – *Journal of Insects as Food and Feed*; eLife - *eLife*. **(B)**. The journal that requires an ethics statement for studies that use invertebrates (*Animal Behaviour*) has a higher mean proportion of papers with an ethics statement compared to journals that do not require an ethics statement for studies that use invertebrates.

of intervention during study design [33]. However, recent studies suggest that methods for determining power or precision often are not used appropriately or transparently reported [34–37]. This can affect the reproducibility and validity of study results, ultimately contributing to translatability and generalizability issues [38]. Increasing the correct use and transparent reporting [1,38] of these methods is an important part of scientific reproducibility and the growing movement towards open research practices [17,39].

### Reporting on methods of sacrifice, anaesthetics, and analgesics

When considering refinements, we looked for reporting that demonstrated anaesthetics were being used prior to invasive handling or death. The majority of papers with invasive handling or death did not report the prior use of analgesics/anaesthetics (84%), despite papers reporting invasive handling and/or death were the vast majority of our sample (88%). Further, when the use of anaesthetics was reported, freezing/chilling was the most frequently reported method (59%; see also [40]). However, invertebrate veterinarians have consistently stated that cold is not considered an appropriate anesthetic for insects prior to invasive procedures due to its lack of analgesic properties and a locomotor confound in determining when unconsciousness has been induced [41,42]. This suggests a lack of guidance from, or interaction with, the veterinary community may be causing entomologists to employ 'anaesthetics' that are not actually recommended as suitable by

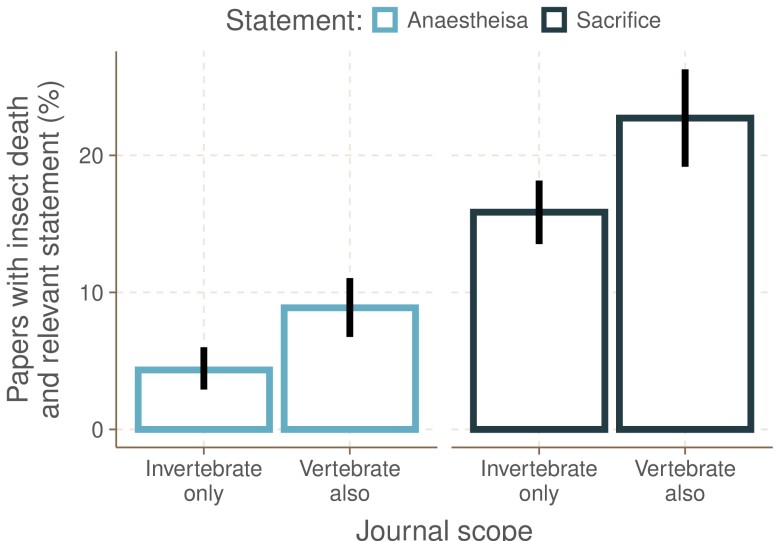

**Fig 4. The percentage of papers reporting relevant anaesthesia or sacrifice statements by journal taxonomic scope.** (blue) Journals that publish vertebrate research as well as invertebrate research had a significantly higher proportion of papers reporting an anaesthesia or analgesia statement compared with journals that publish only invertebrate research ($t$-test, $t_{40,3} = 3.19$, $p < 0.003$). (black) Journals that publish vertebrate research also had a significantly higher proportion of papers reporting a sacrifice statement compared with journals that publish only invertebrate research ($t$-test, $t_{40,38} = 3.05$, $p < 0.005$).

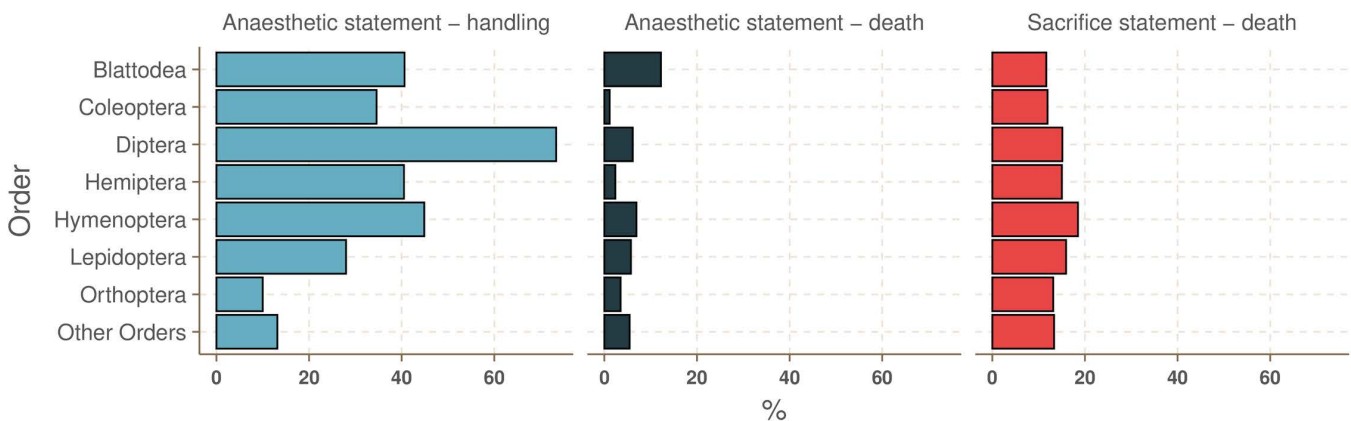

**Fig 5. Percentage of studies with an anaesthetic or sacrifice statement prior to handling or death by order.** Left) Percentage of studies with an anaesthetic statement prior to handling varied by order. Letters indicate statistically significant differences between orders. Middle) There was no significant difference in the percentage of studies with an anaesthetic statement prior to death among orders. Right) There was no significant difference in the percentage of studies with a sacrifice statement prior to death among orders.

veterinarians. This may indicate that allowing self-regulation of welfare within entomology may be insufficient to improve animal welfare in line with modern veterinary guidance without significant changes in how these communities interact. This hypothesis is in line with the judgments of professional entomologists, who have indicated that self-management is likely to be the least effective strategy for protecting insect welfare in research [25], perhaps due to their self-reported lack of knowledge and training on insect welfare.

Importantly, the lack of *reporting* the use of a specific anaesthetic/analgesic agent does not necessarily represent an animal welfare issue in practice, as an anesthetic may still have been used. However, if an anaesthetic was used during the experiment and this was not reported in the methods, this reflects a reproducibility issue because the use of different anaesthetic agents can have both behavioural and physiological consequences for the animal. In *Drosophila*, $CO_2$ anaesthesia can delay copulation [43], shorten lifespans [44], induce metabolic changes [45], and lower fecundity [23,46]. Behavioural impacts include impaired climbing and flight capacity [47]. Other insects are not immune, and the effects of anesthesia can be wide-ranging and specific to a species' unique capacities: for example, $CO_2$ anaesthesia increased bioluminescence in a species of glow-worm [48]. Critically, these effects are not general to all anaesthetics, but can be a result of the specific anaesthetic employed [23]. Thus, not reporting the specific use of an anesthetic agent (or its dose, duration, or exposure method [40]) whenever they *are* employed is a major concern for generating reproducible scientific research.

Anaesthetics may not always be needed to protect an animal's welfare during sacrifice if the sacrifice method employed is instantaneous; this could mean that the low reporting of anaesthetics prior to sacrifice in our studies was simply due to the common usage of instantaneous sacrifice methods. However, many of the most commonly reported sacrifice methods captured in our study were not instantaneous including, most notably, dissection. Dissection as a method of death comprised 26% of our overall study population. Only 17% of papers that reported dissection as a method of death reported the use of an anesthetic or analgesic prior. If this reporting accurately reflects the practices employed by entomologists, vivisection (live, unanaesthetized dissection of an animal) was reported in 24% of studies in our sample that contained insect death (as identified manually) and 20.56% of our overall sample. We found no trend towards increasing reports of anesthesia use over time in studies with dissection as the method of sacrifice.

Importantly, we consider it unlikely that 20.56% of all entomology studies involve vivisection; the inclusion of journals such as *Journal of Experimental Biology, Journal of Insect Physiology,* and *Arthropod Structure and Development* represent disciplines where dissections are more commonly practiced and thus may skew our overall sample of articles in relation to the wider entomological literature. So, while nearly 21% of studies in our sample include vivisection, we should not necessarily think nearly 21% of all entomological research reports the use of vivisection. Nevertheless, this trend in reporting in the journals we have sampled, which are high impact journals that span many entomological subdisciplines, suggests vivisection is common enough to represent a serious animal welfare and public perception concern; vivisection has historically been one of the oldest and most publicly inflammatory animal welfare issues in research [49]. Further, only 35% of the papers that reported the use of any anesthetic used one recognized by veterinary specialists as appropriate for use prior to invasive, surgical, or otherwise stressful/noxious procedures in insects [41,42].

Finally, methods of sacrifice were more frequently reported, with 62% (automated) - 71% (manual) of papers explicitly stating their sacrifice methods. However, just as the method of anaesthesia may affect reproducibility through impacts on physiology, the method of death may also impact study results. For instance, gene expression profiles change as a result of different kinds of stress – well studied, for instance, in the case of heat stress [50]. Plausibly, stress experienced by an insect when frozen instantly in liquid nitrogen is minimal; the stress of drowning [51] in RNAlater may not be. Results from gene expression studies that collect insects using these two different methods thus may not be comparable, which can only be determined through accurate reporting of the sacrifice method employed in each study. This highlights the importance of reporting on sacrifice methods to ensure accurate comparisons across studies can be made and that results can be reproduced.

Many papers also contained a lack of detail around insect death, especially those that trap insects ("samples were collected") or extract nucleic acid ("RNA was extracted from samples following manufacturer's instructions"). The vagaries introduced by use of euphemistic language [52] are an ethical and practical problem that also persists in research using animals covered by legislation; this can undermine the function of reporting methodological information, with detriment impacts on transparency, reproducibility, and reporting on ethical animal use [53].

## Trends over time and by publication

The percentage of lethal studies in our sample increased over time; those containing just invasive handling decreased. Importantly, the percentage of studies that contained an explicit ethics statement to address animal welfare also increased over the study period (2003–2022) from 0% to nearly 8%. Ethics statements in general were rare, and the increase we recorded over time was found to largely be driven by policy changes at the journal *Animal Behaviour,* where ethics statements have been mandated for all studies with invertebrates at least since 2004 (Roth, personal communication).

Even with *Animal Behaviour's* policy change, the vast majority of studies from that journal after 2004 do not contain an ethics statement (> 70%). Poor reporting of welfare relevant information is common, even when studies involve vertebrate animals covered by welfare legislation [53] and not simply journal policies. Top-down enforcement of reporting, from either administrative bodies or journals, is thus only as impactful as the research community enforcing it. Enforcement can be further complicated when both the authors and the reviewers of a community have little training in or guidance on reporting on the ethical treatment of their research animals, as is the case for entomologists [25]. Further, guidance on what to report can vary by the national origin of a publisher or a disciplinary specialty [54], resulting in confusion when authors attempt to prepare publications for submission.

Given that ethics statements are culturally non-standard for entomologists, journals considering implementing policies related to ethics reporting for insects will likely need to 1) create significant resources related to the structure and content of an ethics statement; 2) publicize the change and these resources well in advance of a mandated change [25]; and 3) engage in active enforcement of journal policies at the editorial level, not just the reviewer level, during the transition period.

## Trends by taxonomic scope

Overall, journals that co-published vertebrate research contained a higher proportion of anaesthesia and sacrifice information than those journals that only publish invertebrate research. This suggests that reporting norms may vary across journals, where invertebrate-focused authors that more frequently read and publish in journals that also publish on vertebrates may be slightly more likely to consider ethics reporting even without mandates/enforcement.

Among insect orders, there were no differences in the proportion of sacrifice or anaesthesia statements prior to death. However, there were some differences in the use of anaesthetic statements prior to invasive handling. Studies with Diptera and Hymenoptera had a higher proportion of anesthesia statements than Orthoptera, and studies on Diptera also had a higher proportion of statements than those on Coleoptera, Lepidoptera, and other orders (and see [40]). Interestingly, these results partially confirm the results of [25], where entomologists working with Hymenoptera were more likely to report the use of practices that promote insect welfare than those that did not work with Hymenoptera.

Entomologists working on Diptera and Hymenoptera do not report greater familiarity with, training on, or interest in insect welfare [25], suggesting that these trends in reporting have not been driven by insect-welfare-related reasoning. Increased reporting of anesthetic use due to invasive handling in Hymenoptera could simply be due to occupational health and safety concerns associated with insect stings. Increased reporting in Diptera could be driven by the increased standardization of reporting when using the model organism, *Drosophila melanogaster,* in highly-controlled laboratory settings. Given that the strongest evidence for insect sentience is found in adult Diptera, Blattodea and, to a lesser extent, Hymenoptera [5], it is possible reporting norms in these orders may shift more rapidly moving forward.

Finally, Hemiptera was the fifth-most common order to be studied in our dataset, representing 9% of studies. Despite their popularity as an order for research, no official review of the neurobiological or behavioral data related to Hemipteran sentience has been conducted [5]. Given their heavy use in entomological research, further attention to the neurobiology and behavior of Hemipteran species in relation to the plausibility of their sentience is urgent to better understand our ethical obligations to these animals.

 

## Conclusions

Despite growing concern among entomologists about insect welfare [25], the slow increase in ethics statements over the past twenty years suggests that the field has significant room for improvement, not least to address reproducibility concerns. As importantly, change is likely required to address public concerns related to animal welfare [7], a problem accentuated by the findings of a recent study that showed that 1) trust in scientists declines when the public learns there is an absence of oversight in invertebrate research; and 2) the public believes invertebrates should receive oversight at approximately 2/3rds the level of oversight currently afforded to vertebrates (in Canada [6]). Further, both those arguing for and skeptical of insect sentience agree that some precautionary measures are appropriate to protect welfare when using insects in research [4,7,12,14,15,17,18,55]. Indeed, one of the most-cited articles *critical* of insect sentience ends with the "...recommendation that insects have their nervous systems inactivated prior to traumatizing manipulation…[to guard] against the remaining possibility of pain infliction" [4].

One challenge with changing the entomological community's practice is a lack of resources, training, and guidance to assist community members interested in updating their practices [25]. Entomologists interested in reducing animal use to the minimum number required to have appropriate power can begin by looking at resources for *a priori* analyses [28,29,56]. General guidance is also available from scientific societies on refining research practices when using insects [57] or animals, inclusive of invertebrates [58]; there is also general guidance on insect anesthesia or euthanasia from veterinarians (reviewed in [59] and see [40–42,57])

Entomologists interested in improving their reporting practices, for welfare or reproducibility reasons, should review the ARRIVE guidelines [1], guidance by Spanagel [38], and the requirements of journals like *Animal Behaviour* or *Animal Welfare*, which require more detailed reporting in their ethics statements. Finally, to further facilitate entomologists' reporting of the ethical use of their research animals, we have provided a table (S1 File) that can be modified and included as either a supplement or in the main manuscript, to assist entomologists in capturing reproducibility- and welfare-relevant data.

## Supporting information

**S1 Table. Experimental methods that would result in insect death but were left explicitly unstated within the sampled paper, but could be inferred from reading the methods section.** Some methods were not explicit even on reading methods sections; some animals were "sacrificed" before further experimentation but the exact methods of sacrifice were not stated.
(DOCX)

**S2 Table. Anesthetic methods employed prior to dissection (n = 302 papers).** Not stated category refers to those papers with dissection as the method of sacrifice but no reported anaesthetic or analgesic information.
(DOCX)

**S1 Fig. Number of papers that focused on insects of each order.** Diptera, Hymenoptera, Coleoptera, Lepidoptera, Hemiptera, Orthoptera, and Blattodea were the most represented orders in our analysis. Insecta accounts for papers that reported use of more than 5 orders of insects.
(DOCX)

## Author contributions

**Conceptualization:** R Keating Godfrey, Meghan Barrett.

**Data curation:** Craig D. Perl.

**Formal analysis:** Craig D. Perl.

**Investigation:** Craig D. Perl, Cat Kissinger, Enrique M Castillo.

**Methodology:** Craig D. Perl, Cat Kissinger, R Keating Godfrey, Enrique M Castillo.

**Project administration:** Bob Fischer, Meghan Barrett.

**Supervision:** Bob Fischer, Meghan Barrett.

**Visualization:** Craig D. Perl.

**Writing – original draft:** Craig D. Perl, Meghan Barrett.

**Writing – review & editing:** R Keating Godfrey, Bob Fischer, Meghan Barrett.

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
