## [Decision Letter · Decision Letter 0]

22 Jan 2025

PONE-D-24-41322Identifying trends in reporting on the ethical treatment of insects in researchPLOS ONE

Dear Dr. Barrett,

Thank you for submitting your manuscript to PLOS ONE. After careful consideration, we feel that it has merit but does not fully meet PLOS ONE’s publication criteria as it currently stands. Therefore, we invite you to submit a revised version of the manuscript that addresses the points raised during the review process.

We look forward to receiving your revised manuscript.

Kind regards,

Samuel Adelani Babarinde, PhD

Academic Editor

PLOS ONE

“MB and BF report a relationship with the Insect Welfare Research Society that includes: board of advisors (unpaid). CDP reports a relationship with the Insect Welfare Research Society that includes: employment.”

5. We notice that your supplementary figure and tables are included in the manuscript file. Please remove them and upload them with the file type 'Supporting Information'. Please ensure that each Supporting Information file has a legend listed in the manuscript after the references list.

Reviewers' comments:

Reviewer's Responses to Questions

**Comments to the Author**

1. Is the manuscript technically sound, and do the data support the conclusions?

Reviewer #1: Yes

Reviewer #2: Partly

2. Has the statistical analysis been performed appropriately and rigorously? 

Reviewer #1: Yes

Reviewer #2: No

3. Have the authors made all data underlying the findings in their manuscript fully available?

Reviewer #1: Yes

Reviewer #2: No

4. Is the manuscript presented in an intelligible fashion and written in standard English?

Reviewer #1: Yes

Reviewer #2: No

5. Review Comments to the Author

Reviewer #1: Dear Authors,

thsi study findings is important for the viewpoint about the ethical statement working with invertebrate. but journal name and the publishers name maybe another ethical issue for this paper. maybe you may use categoised group for the publishers or journal name. your results indicated minor slow progress about the ethical arena related to the study areas frocus. for example, dipteran order include general animal and human vectors (Related Arthropod borne disease), therefore any publishers and communities may not want to ethical statement and this is acceptable for the community and publishers related to the reader and beneficiary.

Reviewer #2: Respectfully

The study was statistically and content-wise reviewed.

Please make the following corrections so that the article can be accepted and published in an acceptable manner:

The article "Identifying trends in reporting on the ethical treatment of insects in research" has several notable flaws that could be improved upon:

The study lacks any mention of a priori methods for determining sample sizes (e.g., power analyses), which is essential for reducing the number of animals used in experiments and ensuring ethical rigor.

The majority of studies (84%) did not report the use of appropriate anesthetics or analgesics prior to significant handling or sacrifice. Moreover, freezing/chilling was the most frequently reported method, which is not considered an appropriate anesthetic by veterinary standards.

While 60% of studies explicitly stated their sacrifice methods, the remaining 40% either used vague language or did not provide clear details, undermining transparency and reproducibility.

Although there was an increase in ethics statements over time, they were still included in less than 8% of studies overall. This trend is largely driven by one journal (Animal Behaviour) that mandates ethics reporting.

Reporting practices varied significantly between journals, particularly between those that publish research on vertebrates and invertebrates. Journals that publish vertebrate research showed better reporting on anesthesia and sacrifice practices.

Studies involving Diptera and Hymenoptera were more likely to report the use of anesthetics before invasive handling compared to other insect orders. This discrepancy may reflect inconsistent practices rather than ethical considerations.

The frequent absence of anesthetic use before dissection (83% of dissection studies) raises significant ethical concerns, particularly with practices like vivisection, which can damage public trust.

The lack of detailed reporting on anesthetics, handling methods, and sacrifice techniques creates barriers to reproducibility and comparability of results across studies.

The study's findings about ethics statements are heavily influenced by the journal Animal Behaviour. This overreliance could skew perceptions about trends across the entire field of entomology.

While the article provides tools and recommendations for improving reporting practices, these are not sufficiently robust to address the broader issues of standardization and enforcement.

You can benefit from and cite the following articles to improve the structure and content of your study:

-Knockdown resistance (kdr) associated organochlorine resistance in mosquito-borne diseases (Culex quinquefasciatus): Systematic study of reviews and meta-analysis

-Knockdown resistance (kdr)-associated organochlorine resistance in mosquito-borne diseases (Culex pipiens): A systematic review and meta-analysis

-A review of cultural aspects and barriers to the consumption of edible insects

-Prevalence of Chikungunya, Dengue, and West Nile arboviruses in Iran based on enzyme-linked immunosorbent assay (ELISA): A systematic review and meta-analysis

-Halal Certification for edible insects

The study needs a general revision in terms of language and grammar.

Good luck.

6. PLOS authors have the option to publish the peer review history of their article (what does this mean? ). If published, this will include your full peer review and any attached files.

**Do you want your identity to be public for this peer review?** For information about this choice, including consent withdrawal, please see our Privacy Policy .

Reviewer #1: No

Reviewer #2: **Yes: ** Ebrahim Abbasi

---

## [Author Response · Author response to Decision Letter 1]

14 Apr 2025

Replies to referees are in bold in the cover letter. We've simply copy-pasted here.

Referee 1

Dear Authors,

thsi study findings is important for the viewpoint about the ethical statement working with invertebrate. but journal name and the publishers name maybe another ethical issue for this paper. maybe you may use categoised group for the publishers or journal name. your results indicated minor slow progress about the ethical arena related to the study areas frocus. for example, dipteran order include general animal and human vectors (Related Arthropod borne disease), therefore any publishers and communities may not want to ethical statement and this is acceptable for the community and publishers related to the reader and beneficiary.

We thank the referee for their insightful comment. We are careful throughout to present the ethical and practical benefits of transparent reporting in scientific publication. Many journals and publishers may not want to ask for ethical reporting on insects used in research at this time. We recognize that there is no obligation for them to do so and we do not call for them to introduce any obligatory reporting standards. We only suggest that journals that are considering introducing ethics statements may want to provide clear guidance for authors on the content of those statements, given that our results show ethics statements are not common for entomologists in most journals.

Furthermore, we are careful not to pass any judgement on any journal, paper or author in our study. Open science is a key part of ethical research practices and we provide our raw data in order to adhere to open science principles. We feel that it is essential to be transparent about where we derived our raw data, allowing for greater replication of our analyses and data gathering processes. Journal publishing policies are publicly available, as are all the papers used in our analyses. We have analysed those data but are not revealing anything private or not already in the public domain.

We have used this comment as a prompt to further review our discussion for clarity on the obligation of journals to require reporting on the ethical use of insects in research, making sure to not include any language that suggests we feel journals have this obligation.

Response to Referee 2

In their reply to the yes/no questions, Referee 2 indicates that our methods are partially statistically robust or appropriate.

We have double-checked all our methods and appreciate the referee drawing our attention to re-examine our methodologies. On further reflection, we felt a GLM, quasibinomial model family was more appropriate for analyzing the data in Figure 2 - given the percentage nature of the data and abundance of zeros. Although this does not change any of the conclusions in the manuscript, we have adjusted the statistical analysis and updated our code and the figure to reflect the new analyses.

In their reply to the yes/no questions, Referee 2 indicates that not all data are available for the project.

All data and code are made available at the private for peer review link in the manuscript. If any data are missing, we welcome the referee to share with us which data are not available. As is standard practice with in-review manuscripts, all data will be made publicly available at a non-private version of the same link concurrent with the article’s publication.

Respectfully

The study was statistically and content-wise reviewed.

Please make the following corrections so that the article can be accepted and published in an acceptable manner:

The article "Identifying trends in reporting on the ethical treatment of insects in research" has several notable flaws that could be improved upon:

The study lacks any mention of a priori methods for determining sample sizes (e.g., power analyses), which is essential for reducing the number of animals used in experiments and ensuring ethical rigor.

This study includes no live animals.

The majority of studies (84%) did not report the use of appropriate anesthetics or analgesics prior to significant handling or sacrifice. Moreover, freezing/chilling was the most frequently reported method, which is not considered an appropriate anesthetic by veterinary standards.

While 60% of studies explicitly stated their sacrifice methods, the remaining 40% either used vague language or did not provide clear details, undermining transparency and reproducibility.

Although there was an increase in ethics statements over time, they were still included in less than 8% of studies overall. This trend is largely driven by one journal (Animal Behaviour) that mandates ethics reporting.

Reporting practices varied significantly between journals, particularly between those that publish research on vertebrates and invertebrates. Journals that publish vertebrate research showed better reporting on anesthesia and sacrifice practices.

Studies involving Diptera and Hymenoptera were more likely to report the use of anesthetics before invasive handling compared to other insect orders. This discrepancy may reflect inconsistent practices rather than ethical considerations.

The frequent absence of anesthetic use before dissection (83% of dissection studies) raises significant ethical concerns, particularly with practices like vivisection, which can damage public trust.

The lack of detailed reporting on anesthetics, handling methods, and sacrifice techniques creates barriers to reproducibility and comparability of results across studies.

We appreciate this careful summary of the results of our manuscript and do not see anything to change noted herein.

The study's findings about ethics statements are heavily influenced by the journal Animal Behaviour. This overreliance could skew perceptions about trends across the entire field of entomology.

Referee 2 correctly identifies that conclusions about explicit ethics statements may be skewed by the journal Animal Behaviour. This is something we make clear using data presented in Figures 3 and 4. Further, in Figure 2 we present trends on ethics statements both with and without the inclusion of Animal Behaviour, which we feel does a good job of demonstrating that although Animal Behaviour may have an outsized influence on the numbers and proportion of ethics statements in entomological literature, a similar trend, albeit of a smaller magnitude, does exist even without considering Animal Behaviour. We feel that 3 figures devoted to this topic, alongside accompanying descriptions in the results and discussion text, are sufficient to avoid any confusion and will not unduly skew perceptions across the field of entomology.

While the article provides tools and recommendations for improving reporting practices, these are not sufficiently robust to address the broader issues of standardization and enforcement.

We thank the referee for their thoughtful criticism of the tools and recommendations we present. There are currently no widely available tools or recommendations for improving reporting practices in entomology. We feel that the tools we provide are very good for the purpose for which they are designed, namely, to provide a checklist and materials for researchers to improve their reporting practices.

Further, the aim of this paper is to describe trends in the ethical reporting on insects used in research. Enforcement and standardisation are beyond the scope of our analysis or research question. We leave these important tasks up to the journals, publishers, or legislators that are responsible for processes of enforcement and/or standardisation.

You can benefit from and cite the following articles to improve the structure and content of your study:

-Knockdown resistance (kdr) associated organochlorine resistance in mosquito-borne diseases (Culex quinquefasciatus): Systematic study of reviews and meta-analysis

-Knockdown resistance (kdr)-associated organochlorine resistance in mosquito-borne diseases (Culex pipiens): A systematic review and meta-analysis

-A review of cultural aspects and barriers to the consumption of edible insects

-Prevalence of Chikungunya, Dengue, and West Nile arboviruses in Iran based on enzyme-linked immunosorbent assay (ELISA): A systematic review and meta-analysis

-Halal Certification for edible insects

We thank the referee for the suggestion. Having read the papers in question, it is unclear to us exactly where the referee feels these citations can be best used in this manuscript. We would ask the referee to explain what information those papers contain that are relevant to our manuscript or indicate where in our manuscript they can be best cited; it would help us to understand the specific value these citations will add to our paper. As it stands, this information has not been made clear to us and so we will be forced to politely decline their inclusion as citations.

The study needs a general revision in terms of language and grammar.

We have reviewed our manuscript again for language and grammar and changed any minor typos that we found. The manuscript was written and edited by five native English speakers, two of whom have undergraduate degrees in English and two of whom have graduate degrees in Philosophy. With these native English speakers and individuals with English or Philosophy degrees having reviewed our manuscript, we are satisfied with the language and grammar therein.

---

## [Decision Letter · Decision Letter 1]

14 May 2025

PONE-D-24-41322R1Identifying trends in reporting on the ethical treatment of insects in researchPLOS ONE

Dear Dr. Barrett,

Thank you for submitting your manuscript to PLOS ONE. After careful consideration, we feel that it has merit but does not fully meet PLOS ONE’s publication criteria as it currently stands. Therefore, we invite you to submit a revised version of the manuscript that addresses the points raised during the review process.

We look forward to receiving your revised manuscript.

Kind regards,

Samuel Adelani Babarinde, PhD

Academic Editor

PLOS ONE

Journal Requirements:

Reviewers' comments:

Reviewer's Responses to Questions

**Comments to the Author**

1. If the authors have adequately addressed your comments raised in a previous round of review and you feel that this manuscript is now acceptable for publication, you may indicate that here to bypass the “Comments to the Author” section, enter your conflict of interest statement in the “Confidential to Editor” section, and submit your "Accept" recommendation.

Reviewer #1: All comments have been addressed

Reviewer #3: All comments have been addressed

Reviewer #4: All comments have been addressed

2. Is the manuscript technically sound, and do the data support the conclusions?

Reviewer #1: Yes

Reviewer #3: Yes

Reviewer #4: Yes

3. Has the statistical analysis been performed appropriately and rigorously? 

Reviewer #1: Yes

Reviewer #3: Yes

Reviewer #4: Yes

4. Have the authors made all data underlying the findings in their manuscript fully available?

Reviewer #1: Yes

Reviewer #3: Yes

Reviewer #4: Yes

5. Is the manuscript presented in an intelligible fashion and written in standard English?

Reviewer #1: Yes

Reviewer #3: Yes

Reviewer #4: Yes

6. Review Comments to the Author

Reviewer #1: Thanks for the adequately addressed comments and appropriate answer for the comments. this manuscript may be helpfull fo the future study about the ethical assesments in insects

Reviewer #3: Core Recommendations

1. Enhance transparency in ethical reporting to improve public trust and reproducibility.

2. Develop standardized guidelines for anesthesia, sample size estimation, and sacrifice methods.

3. Foster collaboration between entomologists and veterinarians to align practices with modern welfare standards.

Suggestions for Manuscript Improvement

1. Clarify sampling methodology:

- Specify exclusion/inclusion criteria (e.g., whether studies using pre-killed insects were excluded).

2. Deeper temporal analysis:

- Explore drivers behind increased sacrifice reporting (e.g., journal policy changes vs. researcher awareness).

3. Address ineffective anesthesia:

- Highlight risks of using non-veterinary-approved methods (e.g., chilling/freezing) and their scientific impacts (e.g., altered behavior in Drosophila).

4. Practical tools for researchers:

- Include checklists or templates for ethical reporting in supplements (e.g., ARRIVE guidelines adaptation).

5. Visual enhancements:

- Use distinct colors in figures to differentiate trends across journals or insect orders.

Notable Errors/Improvements

- Writing inconsistencies:

- Table 1 headers are unbalanced (e.g., "Papers containing sacrifice without invasive handling" could be shortened to "Sacrifice Only").

- Standardize terms (e.g., use "anesthesia" instead of "anesthetics/analgesics" in Methods).

- Referencing issues:

- Some citations include broken hyperlinks (e.g., [(Sert et al., 2020)](https://www.zotero.org/google-docs/?7BVuox)). Standardize to APA format.

- Missing figure references (e.g., "Figure 1" and "Figure 2" are cited but not included in the text).

- Language refinement:

- Avoid repetition (e.g., "increase in the proportion of papers reporting insect death" → "rising prevalence of sacrifice reporting").

Reviewer #4: Line 467: please correct the spelling of Lepidoptera

Line 509: I would like to see a few recommendations for proper anesthesia and euthanasia for insects. The manuscript discusses the absence of proper reporting and likely improper handling of insects in experiments, but doesn't provide options on how to improve the problem experimentally. Some entomologists may not know how to properly anesthetize insects.

7. PLOS authors have the option to publish the peer review history of their article (what does this mean? ). If published, this will include your full peer review and any attached files.

**Do you want your identity to be public for this peer review?** For information about this choice, including consent withdrawal, please see our Privacy Policy .

Reviewer #1: No

Reviewer #3: **Yes: ** Nabil ABO KAF

Reviewer #4: No

---

## [Author Response · Author response to Decision Letter 2]

16 May 2025

Reply to reviewers in bold

Journal Requirements:

We don’t believe we’ve cited any retracted papers. Please let us know if that’s not the case.

Reviewer Requests:

Reviewer #1: Thanks for the adequately addressed comments and appropriate answer for the comments. this manuscript may be helpfull fo the future study about the ethical assesments in insects

We thank this reviewer for their comments and were glad to address them and improve the manuscript.

Reviewer #3:

We would like to note, for the editor’s sake and for their editorial process, that we believe the below review was generated by AI given its formatting and incorrect substance (frequently recommending the improvement of things in the manuscript already, suggesting APA formatting, etc.). We are addressing the comments anyway for due diligence, but find addressing incorrect and AI-generated reviewer comments pretty depressing.

Suggestions for Manuscript Improvement

1. Clarify sampling methodology:

- Specify exclusion/inclusion criteria (e.g., whether studies using pre-killed insects were excluded).

We had information about exclusion in lines 143-146. We have added a line about studies with pre-killed insects.

2. Deeper temporal analysis:

- Explore drivers behind increased sacrifice reporting (e.g., journal policy changes vs. researcher awareness).

We have discussed this, insofar as information is available, in lines 449-443. There are no other data to cite on policy changes or researcher awareness.

3. Address ineffective anesthesia:

- Highlight risks of using non-veterinary-approved methods (e.g., chilling/freezing) and their scientific impacts (e.g., altered behavior in Drosophila).

We have addressed this already in lines 369-375.

4. Practical tools for researchers:

- Include checklists or templates for ethical reporting in supplements (e.g., ARRIVE guidelines adaptation).

We have already pointed to the ARRIVE guidelines (line 512) and have a supplemental checklist already made in this manuscript.

5. Visual enhancements:

- Use distinct colors in figures to differentiate trends across journals or insect orders.

Notable Errors/Improvements

- Writing inconsistencies:

- Table 1 headers are unbalanced (e.g., "Papers containing sacrifice without invasive handling" could be shortened to "Sacrifice Only").

We will not make this change, as we want to use consistent terminology in all figures and tables.

- Standardize terms (e.g., use "anesthesia" instead of "anesthetics/analgesics" in Methods)

This would not be a grammatically correct change, therefore we have not made it.

- Referencing issues:

- Some citations include broken hyperlinks (e.g., [(Sert et al., 2020)](https://www.zotero.org/google-docs/?7BVuox)).

Hyperlinks will not be included in the final publication so, as long as the references are complete, we should be okay.

Standardize to APA format.

We have not standardized to APA as that is not the journal’s reference format.

- Missing figure references (e.g., "Figure 1" and "Figure 2" are cited but not included in the text).

Figures mentioned are included in the text; this is not correct.

- Language refinement:

- Avoid repetition (e.g., "increase in the proportion of papers reporting insect death" → "rising prevalence of sacrifice reporting").

We have opted to keep our language precise and consistent, if slightly repetitive, to ensure reader understanding of our results.

Reviewer #4: Line 467: please correct the spelling of Lepidoptera

We have done so.

Line 509: I would like to see a few recommendations for proper anesthesia and euthanasia for insects. The manuscript discusses the absence of proper reporting and likely improper handling of insects in experiments, but doesn't provide options on how to improve the problem experimentally. Some entomologists may not know how to properly anesthetize insects.

We don’t want this paper to turn into a document on how to anesthetize insects - we have already produced, and cited, other guidance in the text that entomologists looking to properly anesthetize their insects can turn to. This is important because different anesthetics, durations and dosages are recommended for different insects, so this is not a simple ‘write one sentence’ fix but rather would require an extensive amount of space be dedicated to insect anesthesia (which is valuable but not the point of the paper). We have instead revised these sentences to make it more clear that the cited references are anesthesia guides FOR insects specifically (and not animals generally): “there is also general guidance on insect anesthesia or euthanasia from veterinarians (reviewed in Barrett, 2024; and see Cooper, 2011; Wahltinez et al., 2023) and in Fischer et al. (2024) and Durosaro and Barrett (2025).”

---

## [Decision Letter · Decision Letter 2]

9 Jul 2025

Identifying trends in reporting on the ethical treatment of insects in research

PONE-D-24-41322R2

Dear Dr. Barrett,

We’re pleased to inform you that your manuscript has been judged scientifically suitable for publication and will be formally accepted for publication once it meets all outstanding technical requirements.

Kind regards,

Salman Khan, Ph.D.

Academic Editor

PLOS ONE

Additional Editor Comments (optional):

Reviewers' comments:

Reviewer's Responses to Questions

**Comments to the Author**

1. If the authors have adequately addressed your comments raised in a previous round of review and you feel that this manuscript is now acceptable for publication, you may indicate that here to bypass the “Comments to the Author” section, enter your conflict of interest statement in the “Confidential to Editor” section, and submit your "Accept" recommendation.

Reviewer #1: All comments have been addressed

Reviewer #3: All comments have been addressed

Reviewer #4: All comments have been addressed

2. Is the manuscript technically sound, and do the data support the conclusions?

Reviewer #1: Yes

Reviewer #3: Yes

Reviewer #4: Yes

3. Has the statistical analysis been performed appropriately and rigorously? 

Reviewer #1: Yes

Reviewer #3: Yes

Reviewer #4: Yes

4. Have the authors made all data underlying the findings in their manuscript fully available?

Reviewer #1: Yes

Reviewer #3: Yes

Reviewer #4: Yes

5. Is the manuscript presented in an intelligible fashion and written in standard English?

Reviewer #1: Yes

Reviewer #3: Yes

Reviewer #4: Yes

6. Review Comments to the Author

Reviewer #1: Dear Authors,

thanks for the adressed all comments and answer adequate for my side. this sentence very important for the ethical statement for the subject area. therefore this manuscript may be helpfull for future aspects for this area

Reviewer #3: The researchers adopted the scientific method in completing the research, which included an appropriate number of samples, appropriate statistical analysis, and sound scientific language.

The evaluation of the first version of the research was assisted by using an artificial intelligence program to save time and invest the potential of modern technologies in the scientific field, especially in the field of evaluation and publication. These were good additions in enriching the evaluation submitted, with appreciation.

Reviewer #4: The authors have addressed my previous comments. I found the article well-written and informative for the scientific audience.

7. PLOS authors have the option to publish the peer review history of their article (what does this mean? ). If published, this will include your full peer review and any attached files.

**Do you want your identity to be public for this peer review?** For information about this choice, including consent withdrawal, please see our Privacy Policy .

Reviewer #1: No

Reviewer #3: **Yes: ** Prof. Dr. Nabil Abo Kaf

Reviewer #4: No

---

## [Editor Report · Acceptance letter]

PONE-D-24-41322R2

PLOS ONE

Dear Dr. Barrett,

I'm pleased to inform you that your manuscript has been deemed suitable for publication in PLOS ONE. Congratulations! Your manuscript is now being handed over to our production team.

Kind regards,

on behalf of

Dr. Salman Khan

Academic Editor

PLOS ONE